# What's New in the Classification, Diagnosis and Therapy of Myeloid Leukemias

**Marco Pizzi [1,*](ID), Carmela Gurrieri [2] and Attilio Orazi [3]**

[1] Surgical Pathology and Cytopathology Unit, Department of Medicine-DIMED, School of Medicine, University of Padua, 35128 Padua, Italy
[2] Hematology and Clinical Immunology Unit, Department of Medicine-DIMED, School of Medicine, University of Padua, 35128 Padua, Italy
[3] Department of Pathology, Texas Tech University Health Sciences Center, El Paso, TX 79905, USA
[*] Correspondence: marco.pizzi.1@unipd.it; Tel.: +39-049-8213794

**Abstract:** Myeloid leukemias are a broad group of hematological disorders, characterized by heterogeneous clinical and biological features. In recent years, unprecedented genetic discoveries and clinical–biological correlations have revolutionized the field of myeloid leukemias. The most relevant changes have specifically occurred in acute myeloid leukemia (AML), chronic myelomonocytic leukemia (CMML), chronic myeloid leukemia (CML) and myeloid neoplasms (MNs) with eosinophilia. The recently published International Consensus Classification (ICC) of myeloid neoplasms has addressed these changes, providing an updated framework and revised diagnostic criteria for such entities. This is also the aim of the 5th edition of the WHO classification of hematopoietic tumors, whose preliminary version was published in 2022. Parallel to this, new therapeutic options and novel molecular targets have changed the management of many myeloid entities, including AML and CML. This review aims to address the most relevant updates in the classification and diagnosis of AML, CMML, CML and MNs with eosinophilia. The state of the art of treatment and future therapeutic options for such disorders are also discussed.

**Keywords:** leukemia; myeloid neoplasms; acute myeloid leukemia; chronic myelomonocytic leukemia; chronic myeloid leukemia; classification; hematopathology

## 1. Introduction

Leukemias are a heterogeneous group of hematologic neoplasms, characterized by clones of circulating myeloid or lymphoid cells. The first description of leukemias as distinct disease entities dates back to the work of J.H. Bennett (1812–1875), R. Virchow (1821–1902) and A.F. Donné (1801–1878) in the mid 1840s [1]. Early classification attempts distinguished two forms of leukemia, namely splenic and lymphatic. The former corresponded to chronic myeloid leukemia (CML), the latter to chronic lymphocytic leukemia/small lymphocytic lymphoma [2]. In 1895, more aggressive disease variants were described and referred to as "acute leukemia" by A. Fränkel (1848–1916) [3]. Finally, in 1913, V. Schilling-Torgau (1883–1960) coined the term "monocytic leukemia" for cases featuring high monocyte counts [4].

Over the twentieth century, the classification of leukemia progressively broadened to include better-defined entities with distinct clinical, morphological, phenotypic and genetic features [5,6]. This integrated approach was endorsed by the 2008 World Health Organization (WHO) classification of hematopoietic neoplasms [7] and by its 2016 revision [8]. Since then, further data have been obtained and two new classifications have recently been published. These are the 2022 International Consensus Classification (ICC) [9] and the 5th edition of the WHO Classification of Hematolymphoid Tumors [10]. Both of them introduced significant changes in the classification and diagnostic criteria of many hematologic neoplasms. Among myeloid leukemias, the most relevant changes regard

acute myeloid leukemia (AML), chronic myelomonocytic leukemia (CMML), CML and myeloid neoplasms (MNs) with eosinophilia.

This review aims at presenting the revised classification and diagnostic criteria of such entities, specifically considering the rationale for changes to prior classifications. For each disease, new therapeutic options and future treatment perspectives are also addressed.

## 2. Updates on the Classification, Diagnosis and Therapy of Acute Myeloid Leukemia (AML)

Over the last few years, the scientific discoveries that occurred in the field of AML have revolutionized our understanding of this neoplasm. High-throughput molecular studies have highlighted new recurrent genetic aberrations, bearing relevant prognostic and therapeutic implications. This, in turn, led to reconsider the classification of AML and to propose new therapeutic approaches.

### 2.1. New Classification and Updates in the Diagnosis of Acute Myeloid Leukemia (AML)

One of the most relevant challenges in the classification of AML is how to integrate clinical, morphological and genetic data to identify non-ambiguous and clinically relevant disease subtypes. The 1976 French–American–British (FAB) classification relied essentially on blast morphology [11]. Since then, phenotypic and molecular studies have demonstrated that AML is highly heterogeneous and that such diversity cannot be captured by morphology alone. This prompted new classification proposals, integrating morphology with clinical, phenotypic and genetic data [6–8,12].

Even this integrated approach, however, poses relevant questions, concerning (i) the type and hierarchy of genetic derangements to include in AML classifications, (ii) the utility of blast thresholds to define specific AML subsets, and (iii) the relevance of former AML subtypes in view of recent genetic acquisitions. All of this was addressed by the 2022 ICC and WHO classifications, which adopted similar (yet not overlapping), genetically oriented approaches [13].

The ICC stratifies AML into five molecular subgroups: (i) AML with recurrent genetic abnormalities (both gene rearrangements and gene mutations); (ii) AML with mutated *TP53*; (iii) AML with myelodysplasia (MDS)-related gene mutations; (iv) AML with MDS-related cytogenetic abnormalities; and (v) AML, not otherwise specified (NOS) (Table 1; Figure 1) [9]. To simplify the classification and avoid confusion among entities with overlapping genetic features, the ICC no longer considers therapy-related AML and AML following MDS, myeloproliferative neoplasms (MPNs) or MDS/MPN as distinct disease subcategories. These, instead, are regarded as diagnostic qualifiers to be added to any AML subtype, whenever appropriate [9,14].

AML with recurrent genetic abnormalities has been expanded to include several newly identified gene rearrangements, as well as AML with *NPM1* mutations and with in-frame bZIP *CEBPA* mutations (Table 1). This substitutes the former AML with biallelic *CEPBA* mutations, since in-frame bZIP derangements have been recently associated with favorable outcome, irrespective of their allelic status [15,16]. No other type of *CEPBA* mutation is included in this category [9]. For all AML with recurrent genetic abnormalities, the minimal blast threshold is now lowered to 10% of nucleated cells, with the notable exception of AML with t(9;22)(q34.1;q11.2), which still requires at least 20% blasts to facilitate its distinction from progression of CML. The 20% threshold also applies to all other AML subgroups (Table 1) [9,14].

**Table 1.** The 2022 ICC and WHO Classification of AML.

| International Consensus Classification (ICC) | | 2022 WHO Classification | |
|---|---|---|---|
| *AML subtypes* | *Blasts \** | *AML subtypes* | *Blasts \** |
| **AML with recurrent genetic abnormalities** | | **AML with defining genetic abnormalities** | |
| Acute promyelocytic leukemia with t(15;17) (q24.1;q21.2)/*PML::RARA* | ≥10% | Acute promyelocytic leukemia with *PML::RARA* fusion | no threshold |
| Acute promyelocytic leukemia with other *RARA* rearrangements | | | |
| AML with t(8;21)(q22;q22.1)/*RUNX1::RUNX1T1* | ≥10% | AML with t(8;21)(q22;q22.1)/*RUNX1::RUNX1T1* fusion | no threshold |
| AML with inv(16)(p13.1;q22) or t(16;16) (p13.1;q22)/*CBFB::MYH11* | ≥10% | AML with *CBFB::MYH11* fusion | no threshold |
| AML with t(9;11)(p21.3;q23.3)/*MLLT3::KTM2A* | ≥10% | AML with *KTM2A* rearrangement | no threshold |
| AML with other *KMT2A* rearrangements | | | |
| AML with t (6;9)(p22.3;q34.1)/*DEK::NUP214* | ≥10% | AML with *DEK::NUP214* fusion | no threshold |
| AML with inv(3)(q21.3q;26.2) or t(3;3)(q21.3;q26.2)/*GATA2::MECOM* | ≥10% | AML with *MECOM* rearrangements | no threshold |
| AML with other *MECOM* rearrangements | | | |
| AML with other rare recurring translocations | ≥10% | AML with other defined genetic alterations | no threshold |
|    AML with t(1;3)(p36.3;q21.3)/*PRDM16::RPN1* | |    AML with *NPM1::MLF1* | |
|    AML with t(3;5)(q25.3;q35.1)/*NPM1::MLF1* | |    AML with *KAT6A::CREBBP* | |
|    AML with t(8;16)(p11.2;p13.3)/*KAT6A::CREBB* | |    AML with *MNX1::ETV6* | |
|    AML with t(1;22)(p13.3;q13.1)/*RBM15::MRTF1* | |    AML with *FUS::ERG* | |
|    AML with t(5;11)(q35.2;p15.4/*NUP98::NSD1* | |    AML with *RUNX1T3(CBFA2T3)::GLIS2* | |
|    AML with t(11;12)(p15.4;p13.3)/*NUP98::KMD5A* | | | |
|    AML with NUP98 and other partners | | | |
|    AML with t(7;12)(q36.3;p13.2)/ETV6::MNX1 | | | |
|    AML with t(10;11)(p12.3;q14.2)/PICALM::MLLT10 | | | |
|    AML with t(16;21)(p11.2;q22.2)/FUS::ERG | | | |
|    AML with t(16;21)(q24.3;q22.1)/RUNX1::CBFA2T3 | | | |
|    AML with inv(16)(p13.3q24.3)/CBFA2T3::GLIS2 | | | |
| AML with t(9;22)(q34.1;q11.2)/*BCR::ABL1* | ≥20% | AML with *BCR:: ABL1* fusion | ≥20% |
| AML with mutated *NPM1* | ≥10% | AML with *NPM1* mutation | no threshold |
| AML with in-frame bZIP *CEBPA* mutations | ≥10% | AML with *CEBPA* mutation | ≥20% |
| **AML with mutated *TP53* \*\*** | ≥20% | - | |
| **AML with myelodysplasia-related gene mutations §** | ≥20% | **AML, myelodysplasia-related** | ≥20% |
| **AML with myelodysplasia-related cytogenetic abnormalities #** | | | |
| **AML, not otherwise specified** | ≥20% | **AML, defined by differentiation** | ≥20% |
| **Myeloid sarcoma** | n.a | **Myeloid sarcoma** | n.a |

Notes: (\*) blast threshold refers to either peripheral blood or bone marrow samples; (\*\*) a separate diagnostic category of AML with mutations in TP53 is not included in the 2022 WHO Classification; (§) defined by mutations in *ASXL1, BCOR, EZH2, RUNX1, SF3B1, SRSF2, STAG2, U2AF1, ZRSR2*; (#) defined by presence of complex karyotype, del(5q)/t(5q)/add(5q), -7/del(7q), +8, del(12p)/t(12p)/add(12p), i(17q), -17/add(17p) or del(17p), del(20q), idic(X)(q13). The WHO category of AML, myelodysplasia related does not include mutations in *RUNX1* and relies on slightly different cytogenetic alterations [9].

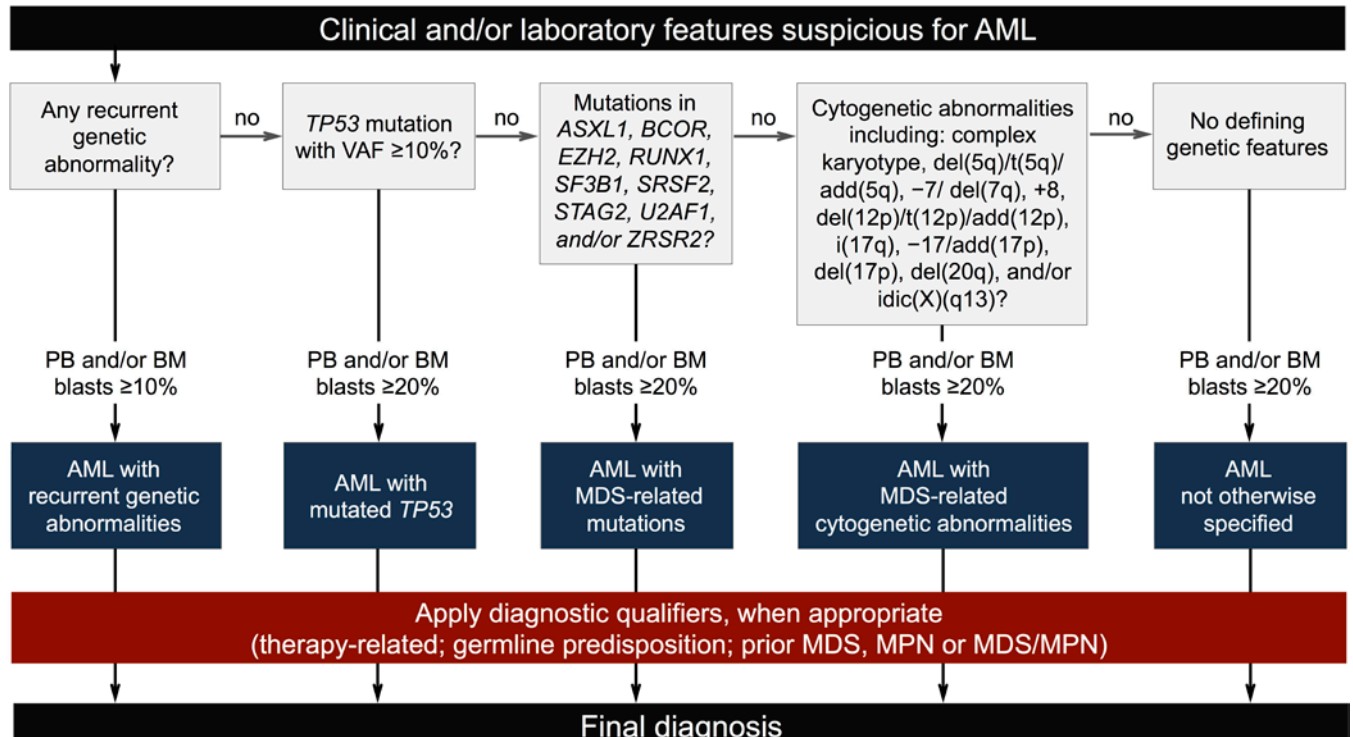

**Figure 1.** The ICC diagnostic flow-chart for AML (modified from [17]).

The newly introduced AML with *TP53* mutation encompasses AML with ≥20% blasts and mono/biallelic mutations of *TP53*, irrespective of blast differentiation and/or phenotype [9]. In all published series, this entity is characterized by particularly poor outcome, transcending both blast count and disease ontogeny (e.g., secondary evolution from prior myeloid neoplasms; therapy relatedness; presence of MDS-related genetic changes) [18–20]. Of note, since most cases of formerly defined pure erythroid leukemia bear *TP53* derangements [21,22], this entity should now be diagnosed as AML with *TP53* mutation, with an optional comment on its erythroid differentiation.

AML with MDS-related cytogenetic abnormalities and AML with MDS-related gene mutations (Table 1) substitute the former category of AML with myelodysplasia-related changes (MRCs) [14]. These new subtypes both require ≥20% blasts. Morphologic dysplasia no longer qualifies an AML as MDS-related, given its poor reproducibility, its limited prognostic value and the occurrence of bone marrow (BM) dysplasia even in cases with no history of MDS (e.g., AML with *NPM1* or *CEPBA* mutations) [23,24]. These subtypes account for about 10% of all AMLs and bear a poor prognosis as previously shown for AML-MRC [25]. Grouping together these molecularly homogenous disorders will likely contribute to a better prognostication and management of patients.

When genetic studies have excluded any of the previously defined molecular abnormalities, a diagnosis of AML, NOS is warranted. In such cases, the type of blast differentiation/morphologic subtype can be specified according to the guidelines outlined in the 4th edition of the WHO Classification. Of note, the number of AML cases diagnosed as NOS is expected to decrease in the near future, as a result of further acquisitions on the molecular bases of these disorders.

In summary, the ICC provides a hierarchical classification of AML, based on genetic determinants with clinical and prognostic relevance. Predisposing factors and/or clinical determinants are downgraded to diagnostic qualifiers that should not impact on the molecular classification (Figure 1).

The 2022 WHO Classification has adopted the same classification framework with a few relevant differences. Unlike the ICC, the WHO does not recognize *TP53* mutations as genetic classifiers and does not consider AML with *TP53* mutations as a separate diagnostic

category [10,13]. Nevertheless, *TP53* derangements are acknowledged as markers of poor prognosis that may impact on therapy decisions. Furthermore, the 2022 WHO Classification (i) includes fewer gene rearrangements in the category of AML with defining genetic abnormalities, (ii) allows any biallelic mutation (not only in-frame bZIP derangements) as well as monoallelic bZIP mutations for the diagnosis of AML with *CEBPA* mutation, (iii) contains slight differences in the list of chromosomal changes and gene mutations defining MDS-related AML and (iv) does not require any blast threshold for the diagnosis of AML with defining genetic abnormalities (with the exception of AML with *BCR::ABL1* fusion and with *CEBPA* mutation) [10] (Figure 2; Table 1).

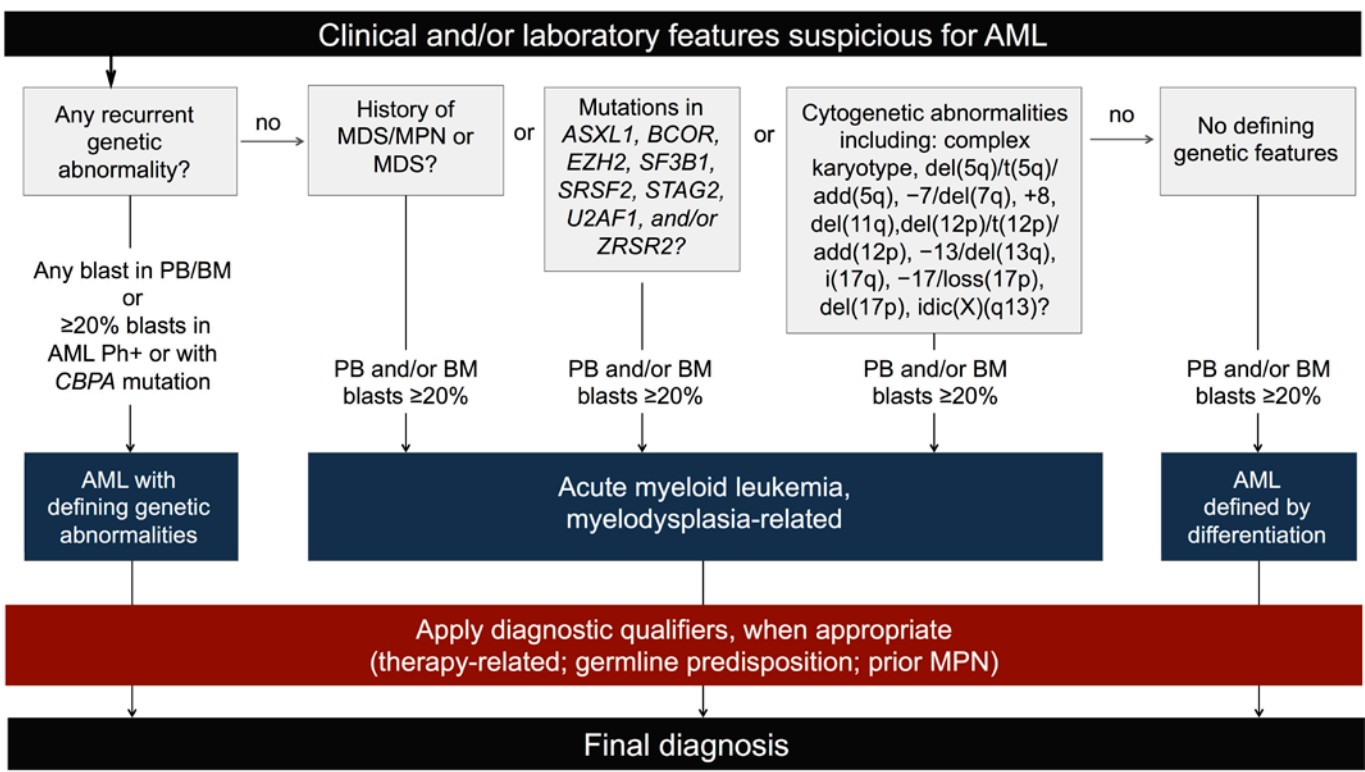

**Figure 2.** The 2022 WHO Classification diagnostic flow-chart for AML.

Therefore, both the ICC and 2022 WHO Classification have significantly revised blast thresholds in genetically defined AML. In detail, while the 2022 WHO Classification regards blast cutoffs as largely arbitrary in this context, the ICC adopts a 10% threshold to support the differential diagnosis with MDS and other myeloid neoplasms, which are still regarded as aggressive AML precursors [9]. Despite these differences, both classifications acknowledge the blurred boundaries between AML and MDS, as illustrated by the extensive revisions that have also been made on MDS classification (Table 2). These include (i) a greater emphasis on genetic derangements, (ii) an update of genetically defined entities (MDS with isolated 5q deletion, with *SF3B1* mutation and with biallelic *TP53* mutations) and (iii) a revised MDS nomenclature, limiting the relevance of morphology. In such a context, the ICC also emphasizes the close relationship between AML and MDS with high blasts (i.e., MDS with 5–19% blasts in PB and/or 10–19% blasts in BM), by renaming these forms MDS/AML and by proposing their molecular stratification according to the AML framework.

**Table 2.** Classification of MDS according to the ICC and 2022 WHO Classification.

| International Consensus Classification (ICC) Criteria | 2022 WHO Classification Criteria |
|---|---|
| MDS with del(5q) | MDS with low blasts and isolated 5q deletion |
| MDS with mutated *SF3B1* | MDS with low blasts and *SF3B1* mutation |
| MDS with mutated *TP53* [a] | MDS with biallelic *TP53* inactivation [a] |
| MDS not otherwise specified (MDS-NOS) [b]<br><br>- MDS-NOS, with single lineage dysplasia<br>- MDS-NOS, with multilineage dysplasia<br>- MDS-NOS, without dysplasia [c] | MDS with low blasts [b]<br><br>- with single lineage dysplasia (*optional*)<br>- with multilineage dysplasia (*optional*) |
| - | MDS, hypoplastic [d] |
| MDS with excess blasts [e] | MDS with increased blasts 1 [e] |
| MDS/AML [f]<br><br>- MDS/AML with MDS-related cytogenetic abnormalities<br>- MDS/AML with MDS-related gene mutations<br>- MDS/AML with mutated *TP53*<br>- MDS/AML not otherwise specified | MDS with increased blasts 2 [f] |
| | MDS with increased blast and fibrosis [g] |

Notes: ([a]) Blast threshold according to 2022 ICC: 0–9% in PB or BM; blast threshold according to 2022 WHO: 0–19% in PB and BM. ([b]) MDS with blasts < 2% blasts in PB and <5% blasts in BM; ([c]) Defined by cytopenia with documented -7, del(7q) or complex karyotype (≥3 independent cytogenetic abnormalities); ([d]) Defined by ≤25% bone marrow cellularity, age adjusted; ([e]) MDS with 2–9% blasts in PB and/or 5–9% in BM; Auer rods also define this entity in 2022 ICC, but exclude it in 2022 WHO; ([f]) MDS with 10–19% blasts in PB and/or BM; Auer rods also define this entity in 2022 WHO, irrespective of blast count; ([g]) MDS with BM fibrosis and 2–19% blasts in PB an/or 5–19% blasts in BM. Abbreviations. AML = acute myeloid leukemia; BM = bone marrow; MDS = myelodysplastic syndrome; PB = peripheral blood.

## 2.2. Novelties in the Therapy of Acute Myeloid Leukemia (AML)

In the second half of the twentieth century, the management of AML sailed on a dead calm sea. Virtually all attempts to identify new drugs failed and treatment of AML basically relied on one-size-fits-all approaches. The only curative option for young and fit patients was induction chemotherapy with cytarabine *plus* an anthracycline ("3 + 7" regimen), followed by consolidation with high-dose cytarabine and, if possible, allogeneic stem cell transplantation (alloSCT) [26]. Older or unfit patients were treated with non-curative regimens, including low-dose cytarabine or, more recently, hypomethylating agents (HMAs) [27,28].

In the last few decades, genetic studies have demonstrated the heterogeneity of AML, laying the foundations for new tailored therapies [6]. In this rapidly evolving scenario, AML risk classification has also changed and will likely keep on changing, due to new molecular markers and to treatment options aimed at neutralizing the poor prognosis of some genetic variants. A paradigmatic example of such developments is the reclassification of *FLT3*-mutated AML as intermediate-risk disease by the 2022 recommendations of the European Leukemia Network (ELN) [17].

In this new era, key issues in AML treatment include: (i) the identification of targetable mutations at diagnosis and relapse; (ii) the choice of appropriate drug combinations to maximize treatment response and to overcome drug resistance; (iii) the choice of low-intensity vs. intensive therapy; (iv) the monitoring of minimal residual disease (MRD) to assess deep remission and relapse risk; (v) the management of new drug-related side effects; and (vi) the appropriate selection of patients for alloSCT to improve transplant outcomes.

Since 2017, several compounds have been approved and many others are being intensively investigated. These comprise both gene mutation-targeting drugs (the *FLT3* inhibitors, midostaurin and gilteritinib; the *IDH1/IDH2* inhibitors, enasidenib, ivosidenib and olutasidenib) and broadly active compounds (the antibody–drug conjugate, gentuzumab

ozogamicin; the liposomal formulation of cytarabine and daunorubicin, CPX-351; the Sonic Hedgehog pathway inhibitor, glasdegib; the *BCL2* inhibitor, venetoclax; the oral azacitidine formulation, CC-486, for maintenance therapy). These can be used as frontline treatment or at relapse [29,30], considering the clinical setting and genomic features of the disease. Gentuzumab ozogamicin is indicated in core binding factor (CBF) AML, CPX-351 in MDS-related and therapy-related AML, midostaurin in *FLT3*-mutated AML and IDH inhibitors in *IDH1/2*-mutated AML [28].

Besides these new compounds, significant advances in so-called low-intensity therapies have been granted by combining HMAs with venetoclax (HMA/V) [31,32]. As an alternative to intensive therapy, HMA/V is characterized by better survival rates and quality of life, especially in old/unfit patients and in relapsing/refractory (R/R) disease [33]. Given its favorable profile, HMA/V has also been investigated as frontline therapy for young adults with high-risk AML, who would otherwise be fit for intensive chemotherapy. Finally, the introduction of CC-486 ushered in a new season for maintenance therapy of AML, as it was associated with improved clinical outcome in patients ineligible for alloSCT, who are initially treated with intensive therapy [34].

Despite this progress, unmet needs remain in the treatment of R/R disease and of AML with adverse genetic features (e.g., AML with *TP53* mutation or *KTM2A* rearrangements) [35,36]. Promising approaches include doublet or triplet drug regimens combining HMA/intensive chemotherapy with venetoclax [31], APR-246 (a mutant p53 reactivator) [37], *FLT3* [38], *IDH1/2* [39] or Menin inhibitors [40], or with monoclonal antibodies against CD47 (magrolimab) [41] and TIM3 (sabatolimab) [42]. Ongoing trials are testing these combinations and will hopefully disclose new ways to treat such aggressive AMLs.

## 3. Updates on the Classification, Diagnosis and Therapy of Chronic Myelomonocytic Leukemia (CMML)

MDS/MPNs are a heterogeneous group of myeloid neoplasms, characterized by both myelodysplastic and myeloproliferative features [43,44]. Over the past decade, significant progress has been made on the molecular bases and natural history of MDS/MPNs. This prompted some relevant changes in the classification and nomenclature of such disorders.

One of the most relevant updates in the ICC classification regards juvenile myelomonocytic leukemia (JMML), a pediatric-age myeloid neoplasm that has been removed from the group of MDS/MPN to be included in a newly created category termed "pediatric and/or germline mutation-associated disorders" [45]. In the 2022 WHO Classification, instead, JMML has been included among the MPNs [10]. Other significnat changes regard MDS/MPN with ring sideroblasts (RSs) and thrombocytosis (MDS/MPN-RS-T). In both the ICC and 2022 WHO Classification, this entity has been renamed MDS/MPN with *SF3B1* mutation and thrombocytosis. The ICC allows diagnosing this condition even in the absence of RSs, provided that *SF3B1* mutations with variant allele frequency (VAF) ≥ 10% are documented [9,44]. The 2022 WHO Classification, instead, requires the documentation of RS in all cases and does not define a minimum VAF for *SF3B1* mutations [10]. Finally, both classifications recognize rare MPD/MPN-RS-T lacking *SF3B1* mutations. The ICC identifies a distinct diagnostic category for such cases (i.e., MDS/MPN-RS-T, NOS) [9], while the 2022 WHO Classification retains them among the MDS/MPNs with *SF3B1* mutation and thrombocytosis, provided that RSs account for ≥15% of erythroid precursors [10].

Besides these differences, the 2022 ICC and WHO Classifications acknowledge recurrent genetic changes, precursor conditions and early disease phases of MDS/MPN that were not accounted for by prior classifications. All of this holds particularly true for CMML, whose diagnostic criteria have been extensively revised to include this new information and to support the differential diagnosis with other entities.

### 3.1. New Classification and Updates in the Diagnosis of Chronic Myelomonocytic Leukemia (CMML)

The ICC proposed a significant revision of CMML diagnostic criteria, specifically focused on (i) lowering the threshold of peripheral blood (PB) monocytosis, (ii) adding

cytopenia as a new diagnostic requirement and (iii) emphasizing the role of genetic abnormalities to assess disease clonality [9]. Furthermore, both the ICC and 2022 WHO Classifications acknowledge the relevance of phenopyic studies to distinguish CMML from other causes of monocytosis (i.e., ≥94% CD14$^+$/CD16$^-$ classical monocytes in PB are strongly associated with CMML as compared to any other condition) [9,10]. In this context, however, it is worth noting that the specificity and (more importantly) sensitivity of flow cytometry assays in CMML have been challenged by recent studies [46,47]. As such, the diagnosis of CMML must always rely on the integration of clinical–pathological, phenotypic and molecular data.

The threshold of PB monocytosis is now set at $0.5 \times 10^9$/L with monocytes being ≥10% of the WBC [9]. This lower value is justified by the results of multi-institutional studies, showing genetic and clinical overlap between conventional CMML (absolute monocytes $\geq 1.0 \times 10^9$/L) and so-called "oligomonocytic" CMML (absolute monocytes $0.5–1.0 \times 10^9$/L) [48,49]. To ensure diagnostic specificity, monocytosis must be accompanied by cytopenia, clonal genetic abnormalities and consistent BM findings (Figure 3). AML and potential CMML mimickers (i.e., MPN, MDS, CML with p190 fusion; myeloid/lymphoid neoplasms with eosinophilia and tyrosine kinase gene fusions (M/LN-Eo-TKs)) must also be excluded. In the absence of genetically defined clonality, the ICC allows a diagnosis of CMML with higher monocyte count (monocytes $\geq 1 \times 10^9$/L and ≥10% of the WBCs) together with morphologic/phenotypic findings (increased PB/BM blasts, morphologic dysplasia and/or CMML-specific phenotypic aberrances by flow cytometry) (Table 3) [9]. In the presence of clonality, the ICC limits the value of morphologic dysplasia, which is now replaced by cytopenia as a marker of ineffective hematopoiesis [44]. This is justified by the poor reproducibility of morphology and by the often-subtle dysplastic changes of most CMMLs. The ICC defines cytopenia as hemoglobin levels <130 g/L (for males) and <120 g/L (for females), absolute PMN count $<1.8 \times 10^9$/L and/or platelet count $<150 \times 10^9$/L [9]. While anemia (with or without thrombocytopenia) is very common in CMML, a small subset of cases lacks cytopenia at diagnosis. These are usually early phase CMML that become cytopenic later in the course of the disease [50].

Based on WBC counts, two disease variants are recognized: CMML with MDS-like phenotype (WBC $< 13 \times 10^9$/L) and CMML with MPN-like phenotype (WBC $\geq 13 \times 10^9$/L) [44]. Distinction between these forms is biologically and clinically relevant, as CMML with MPN-like phenotype is enriched in *RAS* pathway, *JAK*$^{V617F}$ and *SETBP1* mutations and has a more aggressive clinical course [51]. Subgrouping of CMML according to PB/BM blasts is also simplified, by re-adopting a two-tiered system (CMML-1: <5% in PB and <10% in BM; CMML-2: 5–19% in PB and 10–19% in BM). This is justified by the limited prognostic value of distinguishing between prior CMML-0 and CMML-1 subgroups [52,53].

Finally, the ICC recognizes molecular derangements, which correlate with specific clinical–prognostic features (e.g., *NPM1* and *SF3B1* mutations). CMMLs with *NPM1* mutations are associated with particularly poor prognosis and with high risk of AML progression [54,55]. Despite this, *NPM1* mutations should not prompt a diagnosis of AML, if the diagnostic criteria of CMML are met [44]. By contrast, *SF3B1* mutations in cases of otherwise typical CMML are associated with a more favorable outcome with low rates of AML progression. These rare cases present with MDS-like phenotype and are morphologically characterized by numerous RS in the BM [56].

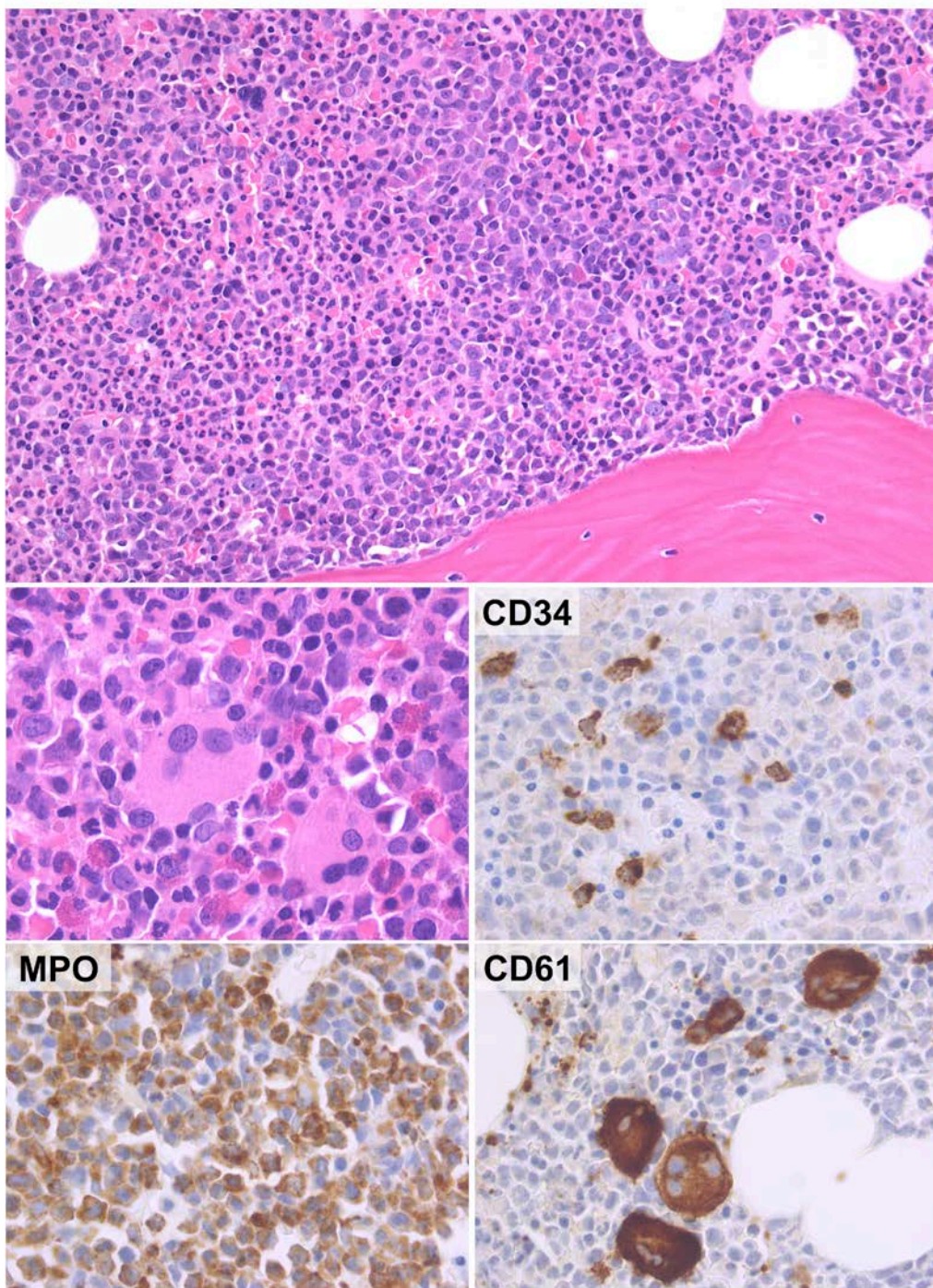

**Figure 3.** Histological features of CMML. The BM of CMML is usually hypercellular with increased, left-shifted granulopoiesis (upper panel). Megakaryocytes are often increased and have MDS-like features (middle left panel). Immunohistochemical analysis discloses variable CD34-positive myeloid precursors and markedly increased MPO-positive myeloid cells. CD61 highlights both atypical megakaryocytes and micromegakaryocytes (H&E and immunoperoxidase stains; original magnification ×20, ×40 and ×63).

**Table 3.** The ICC diagnostic criteria for CMML.

| |
|---|
| Monocytosis defined as monocytes $\geq 0.5 \times 10^9/$L and $\geq 10\%$ of the WBC |
| Cytopenia [a] |
| Blasts (including promonocytes) < 20% of nucleated cells in PB and BM |
| Presence of clonality |
|     Abnormal cytogenetics *and/or* |
|     $\geq 1$ myeloid neoplasm-associated gene mutation (VAF $\geq 10\%$) [b] |
| In cases without evidence of clonality: |
|     monoctyes $\geq 1.0 \times 10^9/$L and $\geq 10\%$ of the WBC with $\geq 1$ of the following |
|      -increased blasts (including promonocytes) [c] |
|      -morphologic dysplasia |
|      -abnormal immunophenotype consistent with CMML |
| BM examination consistent with CMML (hypercellularity due to myeloid proliferation often with increased monocytes) and lacking diagnostic features of AML, MPN or other conditions associated with monocytosis |
| No *BCR::ABL1* fusion or genetic abnormalities consistent with M/LN-eo-TK |

Notes: ([a]) rare cases may show borderline or no cytopenia usually in phase disease; ([b]) based on International Consensus Group Conference, Vienna, 2018 [48]; ([c]) defined as blasts $\geq 5\%$ in BM and $\geq 2\%$ in PB. Abbreviations. AML = acute myeloid leukemia; BM = bone marrow; CMML = chronic myelomonocytic leukemia; M/LN-eo-TK = myeloid/lymphoid neoplasms with eosinophilia and tyrosine kinase gene fusions; MPN = myeloproliferative neoplasm; PB = peripheral blood; WBC = white blood cells.

Of note, the ICC also introduced clonal monocytosis of undetermined significance (CMUS) as a precursor condition of CMML. CMUS is defined by persistent monocytosis (monocytes $\geq 0.5 \times 10^9/$L and $\geq 10\%$ of the WBC) with at least one myeloid neoplasm-associated mutation (variant allele frequency $\geq 2\%$) and with BM findings inconsistent with CMML or other types of myeloid malignancy (Table 4) [9]. If cytopenia is also present, a diagnosis of clonal cytopenia and monocytosis of undetermined significance (CCMUS) is warranted [9]. CMUS and CCMUS must be distinguished from reactive monocytosis, since they frequently progress to overt CMML and other myeloid malignancies [57]. Thus, a thorough diagnostic workup is always recommended before making a diagnosis of CMUS or CCMUS.

The 2022 WHO Classification introduces several changes also adopted by the ICC (Table 5), but it proposes a hierarchical approach for the diagnosis of CMML with slight differences from the ICC scheme. In particular, the 2022 WHO Classification does not include cytopenia as a diagnostic requirement for CMML and retains morphologic dysplasia as a necessary diagnostic criterion for cases with low monocyte counts ($0.5$–$1 \times 10^9/$L) [10]. Moreover (in contrast to the ICC approach), the WHO 2022 Classification mandates to diagnose all *NPM1*-mutated myeloid neoplasms with CMML features as AML with *NPM1* mutation, given their high rate of acute leukemic progression. Finally, the 2022 WHO Classification does not recognize CMUS and CCMUS as separate diagnostic categories [10].

**Table 4.** The ICC diagnostic criteria for CMUS.

| |
|---|
| Persistent monocytosis defined as monocytes $\geq 0.5 \times 10^9/$L and $\geq 10\%$ of the WBC |
| Absence or presence of cytopenia [a] |
| Presence of $\geq 1$ myeloid neoplasm-associated gene mutation (VAF $\geq 2\%$) [b] |
| No significant dysplasia, increased blasts (including promonocytes) or morphologic findings of CMML on BM examination [c] |
| No criteria for a myeloid or other hematopoietic neoplasm are fulfilled |
| No reactive conditions that would explain monocytosis are detected |

Notes: ([a]) if cytopenia is present, the nomenclature of clonal cytopenia and monocytosis of undetermined significance (CCMUS) is suggested; ([b]) based on International Consensus Group Conference, Vienna, 2018 [48]; ([c]) BM findings of CMML include hypercellularity due to myeloid proliferation often with increased monocytes and (in subsets of cases) monoblasts and/or blast equivalents (i.e., promonocytes) and/or dysplasia in $\geq 1$ lineage. Abbreviations. BM = bone marrow; CMML = chronic myelomonocytic leukemia; CMUS = clonal monocytosis of undetermined significance VAF = variant allele frequency; WBC = white blood cells.

**Table 5.** The 2022 WHO diagnostic criteria for CMML.

| Prerequisite Criteria |
| --- |
| 1. Persistent monocytosis, defined as monocytes $\geq 0.5 \times 10^9$/L and $\geq 10\%$ of the WBC |
| 2. Blasts [a] < 20% of nucleated cells in PB and BM |
| 3. Not meeting diagnostic criteria of CML or other MPN |
| 4. Not meeting diagnostic criteria of M/LN-eo-TK |
| **Supporting Criteria** |
| 1. Dysplasia involving $\geq 1$ myeloid lineage [b] |
| 2. Acquired clonal cytogenetic or molecular abnormality |
| 3. Abnormal partitioning of PB monocyte subsets [c] |

| Diagnostic Requirements |
| --- |
| A diagnosis of CMML is posed if all prerequisite criteria are present *together with*: |
| (a)   $\geq 1$ supporting criteria, if monocytosis is $\geq 1 \times 10^9$/L |
| (b)   both supporting criteria #1 and #2, if monocytosis is $0.5 - 1.0 \times 10^9$/L |

| Subtyping Criteria |
| --- |
| - Myelodysplastic CMML: WBC $< 13 \times 10^9$/L |
| - Myeloproliferative CMML: WBC $\geq 13 \times 10^9$/L |
| **Subgrouping Criteria** (based on percentage of blasts and promonocytes) |
| - CMML-1: <5% in PB and <10% in BM |
| - CMML-2: 5–19% in PB and 10–19% in BM |

Notes: ([a]) blast count includes myeloblasts, monoblast and promonocytes; ([b]) dysplasia should be present in $\geq 10\%$ of cells of a hematopoietic lineage in the BM; ([c]) based on detection of >94% classical monocytes, in the absence of known active autoimmune disease and/or systemic inflammatory syndromes. Abbreviations: BM = bone marrow; CML = chronic myeloid leukemia; CMML = chronic myelomonocytic leukemia; M/LN-eo-TK = myeloid/lymphoid neoplasms with eosinophilia and tyrosine kinase gene fusions; MPN = myeloproliferative neoplasm; PB = peripheral blood; VAF = variant allele frequency; WBC = white blood cells.

### 3.2. Novelties in the Therapy of Chronic Myelomonocytic Leukemia (CMML)

Unlike other myeloid malignancies, the treatment of CMML remains challenging. This is largely due to the biological heterogeneity and intrinsic chemoresistance of the disease and to the lack of dedicated clinical trials for such a rare entity [58]. To date, the only curative approach for CMML is alloSCT, but this is indicated in only a minority of cases, due to the advanced age and comorbidities of many patients [58,59]. As such, treatment options are largely confined to controlling myeloid proliferation in cases with MPN-like phenotype and to compensating cytopenia in cases with MDS-like phenotype.

As for the control of myeloid proliferation, seminal studies in the late 1990s reported disappointing results for chemotherapy, with minimal (if any) clinical response and high toxicity rates [60]. Since then, hydroxyurea (HU) has been increasingly used to control symptoms and disease burden. More recently, HMAs have also been associated with satisfactory response and good safety profiles. The recommended HMAs in CMML include 5-azacitidine (5-AZA), decitabine and the oral formulation ASTX727 (decitabine/cedazuridine). Of note, these recommendations are based on clinical trials designed mostly for MDS and including only small subsets of CMML [61–64].

Later real-world studies and small non-randomized clinical trials on HMA-treated CMML reported 40–50% overall response (OR) and 7–17% complete response (CR) rates [65–67]. However, HMA had no effect on mutational allele burden and on acute leukemic progression, even in patients undergoing clinical response [68,69]. Furthermore, a large retrospective multicenter study suggested a survival advantage for HMA-treated high-risk myeloproliferative CMML, with minimal (if any) effect on lower-risk disease [70]. Finally, a recent phase 3 clinical trial (DACOTA trial) failed to demonstrate any survival advantage for HMAs over HU [71]. As such, no conclusion can be drawn on the best treatment for myeloid proliferation in CMML and both HMAs and HU are the standard of care in this setting.

In recent years, new therapeutic opportunities for CMML have been identified. These include (i) combination therapies with HMA (e.g., 5-AZA *plus* venetoclax; 5-AZA *plus* IDH1/2 inhibitors), (ii) new compounds for CMML-related cytopenias (e.g., sotatercept or luspatercept for anemia; eltrombopag for thrombocytopenia) and (iii) therapies targeting

CMML-associated molecular derangements (e.g., inhibitors of GM-CSF, RAS-MAPK or Sonic Hedgehog pathways) [58]. Unfortunately, none of these is curative and further research is needed to address the many unmet needs of CMML therapy.

## 4. Updates on the Classification, Diagnosis and Therapy of Chronic Myeloid Leukemia (CML)

Since the publication of the 2016 WHO Classification, clinical and molecular advances have prompted relevant changes in phase stratification of CML. Significant updates have specifically focused on accelerated phase (AP) and blast phase (BP) disease. This has been paralleled by the advent of new therapeutic strategies, aimed at overcoming therapy resistance to conventional tyrosine kinase inhibitors (TKIs).

### 4.1. New Classification and Updates in the Diagnosis of Chronic Myeloid Leukemia (CML)

The ICC retains the traditional partition of CML into three phases and proposes simplified criteria for the diagnosis of AP and BP disease [9]. AP is now defined by (i) 10–19% blasts in PB and/or BM, (ii) $\geq$20% basophils in PB and/or (iii) additional clonal cytogenetic abnormalities in the neoplastic clone (i.e., second Philadelphia chromosome; trisomy 8; isochromosome 17q; trisomy 19; abnormalities of 3q16.2; complex karyotype, defined as $\geq$3 cytogenetic abnormalities) (Table 6) [9,72]. The 2016 criteria of worsening thrombocytopenia and increasing leukocytosis, thrombocytosis or splenomegaly unresponsive to therapy have been dropped from the definition of AP. Likewise, the provisional criteria of response to TKIs are now excluded from the defining features of AP [8,9]. In line with the 2016 WHO Classification, the ICC acknowledges that BM fibrosis often associated with increased, dysplastic megakaryopoiesis is common in AP [73] (Figure 4), but this is not a stand-alone criterion for the diagnosis of disease progression [72]. The ICC also better specifies the diagnostic criteria of BP, which now include (i) $\geq$20% blasts in PB and/or BM, (ii) myeloid sarcoma (i.e., extra-medullary blast proliferation) or (iii) >5% lymphoid blasts in PB and/or BM (i.e., lymphoblastic crisis) (Table 6) [72,74].

**Table 6.** ICC diagnostic criteria of accelerated and blast phase CML.

| **Diagnostic Criteria of Accelerated Phase CML** *(Any of the Following)* |
|---|
| - 10–19% blasts in BM or PB |
| - $\geq$20% basophils in PB |
| - Additional clonal cytogenetic abnormalities in Philadelphia-positive cells [a] |
| **Diagnostic criteria of blast phase CML** *(any of the following)* |
| - $\geq$20% blasts in BM or PB |
| - Myeloid sarcoma [b] |
| - >5% morphologically apparent lymphoblasts warrants consideration of lymphoblastic crisis |

Notes: ([a]) Additional clonal cytogenetic abnormalities include: second Philadelphia chromosome, +8, iso17q, +19, complex karyotype, or abnormalities of 3q26.2. ([b]) defined as extramedullary blast proliferation. Abbreviations: BM = bone marrow; CML = chronic myeloid leukemia; PB = peripheral blood.

The 2022 WHO Classification adopts an alternative, biphasic scheme for the natural history of CML, only recognizing chronic phase (CP) and BP disease. AP is no longer considered an independent step in the biological continuum of CML and is regarded as "high-risk CP" [10]. The removal of AP as a diagnostic category is paralleled by the introduction of a list of features that correlate with high-risk CP (Table 7) [10,75]. All the ICC defining parameters of AP are included in it [9,72]. Finally, the 2022 WHO criteria of BP largely correspond to those of the ICC, although no actual threshold for lymphoblastsis provided by the WHO [10]. In conclusion, despite differences in terminology, the 2022 ICC and WHO Classifications provide roughly similar risk stratifications for CML, which are in keeping with clinical observations following the introduction of TKIs.

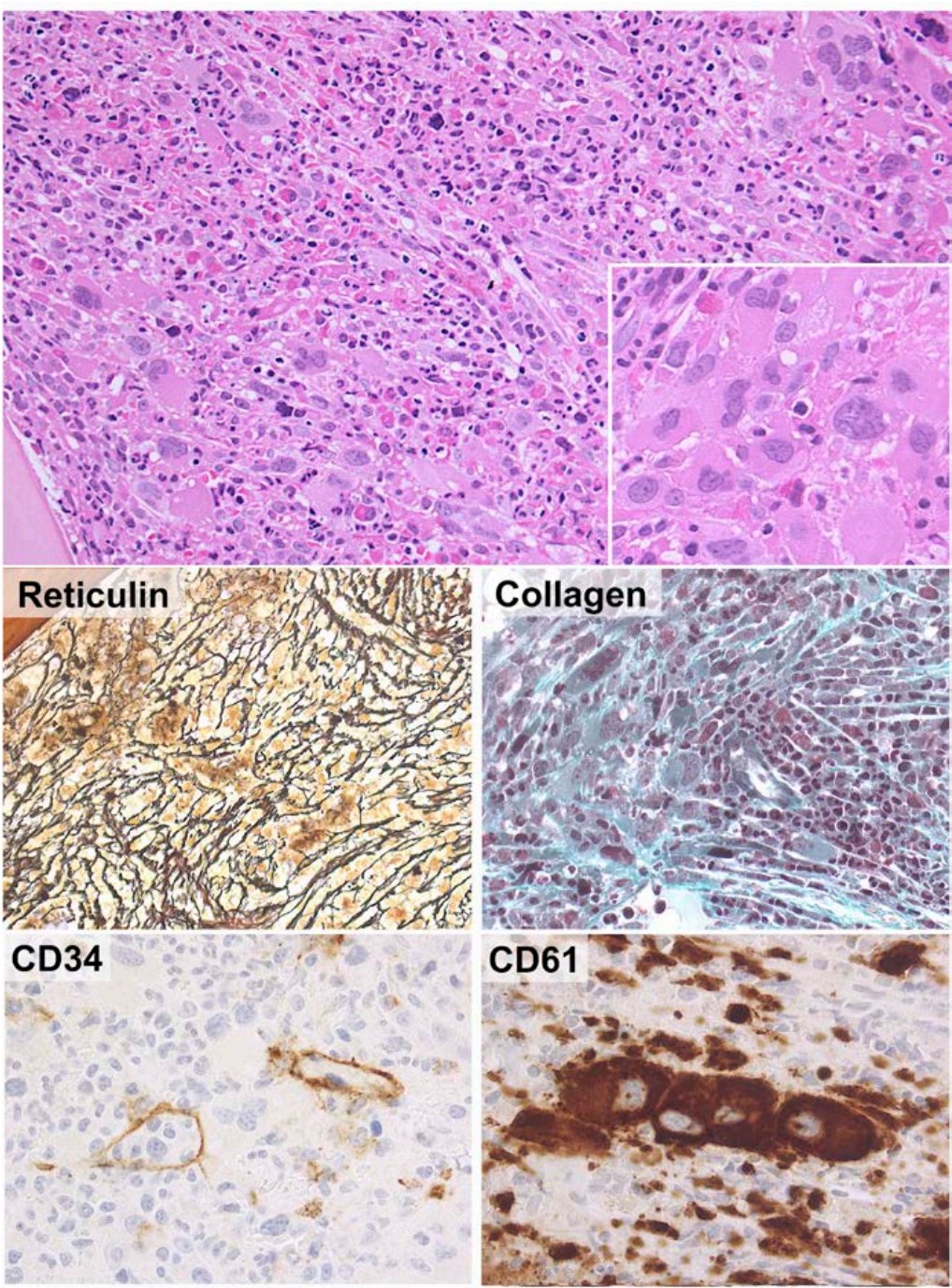

**Figure 4.** BM histological features correlating with high-risk CML. In rare cases, the BM of CML shows markedly increased megakaryocytopoiesis with clusters of atypical (MDS-like or myelofibrosis-like) megakaryocytes (upper panel, insert). These findings are usually associated with severe reticulin and collagen fibrosis (middle panels). Immunohistochemical analysis for CD34 reveals patent sinusoids with intra-vascular hematopoiesis. Increased CD34-positive myeloid precursors may or may not be documented. CD61 immunostain is positive in tight clusters of megakaryocytes. These histological features are associated with a more aggressive clinical course and poorer response to TKIs (H&E and immunoperoxidase stains; original magnification ×20, ×40 and ×63).

**Table 7.** Features associated with increased risk of progression in chronic phase CML according to the 2022 WHO Classification.

| **At Diagnosis** *(Any of the Following)* |
| --- |
| - High ELTS score |
| - 10–19% blasts in BM and/or PB [a] |
| - ≥20% basophils in PB |
| - Additional clonal cytogenetic abnormalities in Philadelphia-positive cells [b] |
| - Clusters of small megakaryocytes (including micromegakaryocytes) with increased reticulin and/or collagen fibrosis. |
| **Emerging on treatment** *(any of the following)* |
| - Failure to achieve a complete hematological response to the first TKI |
| - Any indication of resistance to 2 sequential TKIs (excluding explicable causes, such as kinase domain mutations resistant to any of the administered TKIs) |
| - Development of new additional chromosomal abnormalities |
| - Occurrence of compound mutations in the *BCR::ABL1* fusion gene during TKI therapy |

Notes: ([a]) the finding of lymphoblasts in PB or BM (even if <10%) is consistent with blast phase disease; ([b]) Additional clonal cytogenetic abnormalities include: second Philadelphia chromosome, -7, iso17q, complex karyotype, or 3q26.2 rearrangements. Abbreviations. BM = bone marrow; ELTS = European Treatment Outcome Study (EUTOS) long-term survival [75]; PB = peripheral blood; TKI = tyrosine kinase inhibitors.

### 4.2. Novelties in the Therapy of Chronic Myeloid Leukemia (CML)

The discovery of the Philadelphia chromosome paved the way to TKI-based therapies for CML [76]. This has dramatically changed the paradigm of CML management, prompting deep and fast molecular responses, decreasing disease progression and prolonging life expectancy to nearly that of the general population. Long-term outcome studies specifically indicate that TKIs have increased the 10-year OS of CML from roughly 20% to 80–90% [77,78].

Several TKIs are available for the treatment of CML, including imatinib (first-generation TKIs, 1G-TKIs), dasatinib, nilotinib and bosutinib (second-generation TKIs, 2G-TKIs), ponatinib, olverembatinib and vodobatinib (third-generation TKIs) and PF-114 (fourth-generation TKI). While 1G-TKI, 2G-TKI and ponatinib have entered clinical practice [79–82], olverembatinib, vodobatinib and PF-114 are still under investigation in dedicated trials [83]. In October 2021, a new allosteric inhibitor of BCR-ABL1 (asciminib) joined the TKI family [84]. Unlike other compounds, asciminib binds the myristoyl pocket of BCR-ABL1, blocking its kinase activtability. This unique mechanism may prove particularly useful to overcome resistance to conventional TKIs.

To date, key points in the management of CML include: (i) the choice of first-line TKI; (ii) the management of TKI resistance and side effects; (iii) the prevention and treatment of AP/BP disease; (iv) therapy withdrawal following remission (aka therapy-free remission, TFR).

As for the choice of upfront therapy, current guidelines endorse any 1G/2G-TKI as a therapeutic option [85]. The choice usually relies on several parameters, including disease risk stratification, patient comorbidities, treatment goals and costs and physician and patient preference [86]. Patients with intolerance/resistance to 1G/2G-TKIs may receive a further 2G-TKI or ponatinib/asciminib, after careful re-assessment of disease phase and clonal evolution. AlloSCT still represents the only curative treatment for CP disease resistant to ≥2 TKIs. As for BP disease, the therapeutic options include AML-like or B-ALL-like regimens (depending on blast phenotype) in addition to TKIs. AlloSCT also represents a valuable therapeutic option in all elegible patients, with 5-year overall survival rates of about 60% [87].

Another major issue in the therapy of CML is when and how to consider TKI discontinuation. Starting from the STIM1 study [88] and from reports on interferon (IFN)-treated CML [89], several trials and real-world studies have demonstrated the safety of TFR in selected patients with CP CML. This has led to incorporation of TFR as a goal of treatment by the 2020 ELN and National Comprehensive Cancer Network (NCCN) recommendations [90,91].

Factors that are associated with TFR include (i) the baseline disease risk category, (ii) the duration of TKI treatment, (iii) the response to TKI and (iv) the depth and duration of molecular response [92]. Since only a minority of patients achieves durable TFR, many strategies are under investigation to increase these figures. Possible options include asciminib as frontline therapy (NCT05143840 trial), combination therapies with ruxolitinib and TKIs (NCT03610971 trial) and further administration of TKIs following TFR failure (NCT04838041 trial). The results of these studies will hopefully improve TFR rates, marking further progress in the management of CML.

**5. Updates on the Classification, Diagnosis and Therapy of Myeloid Neoplasms (MNs) with Eosinophilia**

MNs with eosinophilia are a heterogeneous group of myeloid disorders that mainly include chronic eosinophilic leukemia, NOS (CEL, NOS) and the renamed, genetically assigned group of M/LN-Eo-TK. The diagnostic criteria for CEL, NOS have been updated and its boundaries with other eosinophilic disorders have been refined, *plus* new entities have been added to the group of M/LN-Eo-TK. These changes have a direct impact on the management of patients.

*5.1. New Classification and Updates in the Diagnosis of Myeloid Neoplasms (MNs) with Eosinophilia*

The 2022 ICC and WHO Classifications introduced significant changes to the diagnostic criteria of CEL, NOS (Table 8) [9]. In particular, the ICC adopts a more strict definition of hypereosinophila (HE) and introduces BM morphology as a supportive diagnostic criterion. According to the ICC, HE is now defined by high absolute eosinophil counts ($\geq 1.5 \times 10^9$/L) with relative eosinophilia ($\geq 10\%$ of the WBCs). This is in line with the diagnostic approach to other MNs (e.g., monocyte count in CMML) and avoids ambiguities in cases with extreme leukocytosis and high absolute eosinophil counts but low percentage of eosinophils [93]. In recent years, BM morphology has emerged as a robust diagnostic marker of CEL, NOS, since marrow dysplasia and/or reticulin fibrosis are typical of this condition and almost never observed in non-clonal HE [94]. As such, in the presence of persistent eosinophilia, diagnostically appropriate BM findings can be usedto diagnose CEL, NOS even in the absence of clonal abnormalities and/or increased blasts, after other possible causes of eosinophila have been excluded (Table 8) [9].

**Table 8.** Diagnostic criteria for CEL, NOS according to the ICC and 2022 WHO Classification.

| International Consensus Classification (ICC) Criteria | 2022 WHO Classification Criteria |
|---|---|
| Eosinophilia defined as eosinophils $\geq 1.5 \times 10^9$/L and $\geq 10\%$ of the WBC | Eosinophilia defined as eosinophils $>1.5 \times 10^9$/L on at least 2 occasions over an interval of at least 4 weeks |
| Blasts < 20% of nucleated cells in PB and BM, not meeting any other diagnostic criteria for other AML<br>No tyrosine kinase fusions including *BCR::ABL1*, other *ABL1*, *PDGFRA*, *PDGFRB*, *FGFR1*, *JAK2*, *FLT3* fusions<br>Not meeting criteria for other well-defined MPN, CMML or SM [a] | Not meeting criteria for any other myeloid or lymphoid |
| BM examination showing increased cellularity and dysplastic megakaryocytes with/without dysplastic features in other lineages and often significant fibrosis, associated with eosinophilic infiltrate *or*<br>$\geq 5\%$ blasts in the BM and/or $\geq 2\%$ blasts in the PB | BM examination showing increased cellularity and dysplastic megakaryocytes with/without dysplastic features in other lineages and often significant fibrosis, associated with eosinophilic infiltrate |
| Demonstration of a clonal cytogenetic abnormality and/or somatic mutation [b] | Demonstration of a clonal cytogenetic abnormality and/or somatic mutation [b] |

Notes: ([a]) CEL, NOS may occur as SM with associated myeloid neoplasm; ([b]) In the absence of clonal cytogenetic abnormalities/somatic mutations or increased blasts, the ICC allows a diagnosis of CEL, NOS if consistent BM findings and persistent eosinophilia are documented and all other causes of eosinophilia are excluded. This is not allowed by the 2022 WHO Classification, which mandates clonal genetic abnormalities in all cases. Abbreviations. AML = acute myeloid leukemia; BM = bone marrow; CMML = chronic myelomonocytic leukemia; MPN = myeloproliferative neoplasm; PB = peripheral blood; SM = Systemic Mastocytosis; WBC = white blood cells.

The 2022 WHO Classification adopts slightly different diagnostic criteria, as it (i) requires only high absolute eosinophil counts ($\geq 1.5 \times 10^9$/L) with no relative eosinophilia, (ii) considers both clonal genetic derangements and abnormal BM findings as mandatory features and (iii) dismisses high blast counts from the diagnostic checklist (Table 8) [10].

As for other MNs with eosinophilia, both the 2022 ICC and WHO Classifications have renamed the category of M/LN-Eo-TK to emphasize the relevance of TK gene fusions in the pathogenesis and therapy of such conditions. Furthermore, the former list of M/LN-Eo-TKs (i.e., M/LN with *PDGFRA*, *PDGFRB* and *FGFR1* rearrangements) has been expanded to also include M/LN with *FLT3* and *ETV6::ABL1* gene fusions as well as M/LN with *JAK2* rearrangements other than those with *PCM1* [9,10,93] (Table 9). The WHO Classification has introduced a further diagnostic category of "M/LN with other TK rearrangements" for entities presenting as M/LN-Eo-TK with rare or unconventional genetic features [10] (Table 9).

**Table 9.** Classification of M/LN-Eo-TK according to the ICC and 2022 WHO Classification.

| International Consensus Classification (ICC) | 2022 WHO Classification |
|---|---|
| M/LN with *PDGFRA* rearrangement [a] | M/LN with *PDGFRA* rearrangement [a] |
| M/LN with *PDGFRB* rearrangement [b] | M/LN with *PDGFRB* rearrangement [b] |
| M/LN with *FGFR1* rearrangement [c] | M/LN with *FGFR1* rearrangement [c] |
| M/LN with *JAK2* rearrangement [d] | M/LN with *JAK2* rearrangement [d] |
| M/LN with *FLT3* rearrangement [e] | M/LN with *FLT3* rearrangement [e] |
| M/LN with *ETV6::ABL1* | M/LN with *ETV6::ABL1* rearrangement |
| | M/LN with other TK fusions (*ETV6::FGFR2* fusion; *ETV6::LYN* fusion; *ETV6::NTRK3* fusion; *RANBP2::ALK* fusion; *BCR::RET* fusion; *FGFR1OP::RET* fusion) |

Notes ([a]) most common fusion: *FIP1L1::PDGFRA*; other partner genes: *CDK5RAP2, STRN, KIF5B, TNKS2, ETV6, BCR*; ([b]) most common fusion: *ETV6::PDGFRB, but >30 alternative partners reported*; ([c]) most common fusion: *ZMYM2::FGFR1*, but >15 alternative partners reported (including BCR); ([d]) most common fusion: *PCM1::JAK2*; other partner genes: ETV6 and BCR ([e]) most common fusion: *ETV6::FLT3*; other partner genes: *ZMYM2, TRIP11, SPTBN1, GOLGB1, CCDC88C, MYO18A, BCR*; Abbreviations. M/LN = myeloid/lymphoid neoplasm; M/LN-Eo-TK = myeloid/lymphoid neoplasm with eosinophilia and tyrosine kinase gene fusions.

M/LN-Eo-TKs have very heterogeneous clinical–pathological presentation, resembling either MPN, MDS/MPN, AML or B-cell and T-cell acute lymphoblastic leukemia (ALL). In this latter case, to avoid confusion with conventional ALL with TK rearrangements, a diagnosis of M/LN-Eo-TK is appropriate if (i) there is concurrent or previous evidence of an MN, (ii) a residual MN is documented after ALL therapy or (iii) TK rearrangements are found in myeloid cells [93]. Likewise, since mast cell clones resembling systemic mastocytosis (SM) can be documented in M/LN-Eo-TK, assessment for TK rearrangements is recommended in all SM with HE or lacking *KIT* mutations. If TK rearrangements are documented, the diagnosis of M/LN-Eo-TK supersedes that of SM [93].

*5.2. Novelties in the Therapy of Myeloid Neoplasms (MNs) with Eosinophilia*

The treatment of MNs with eosinophilia is challenging, given the limited number of studies on this topic and the overall rarity of such conditions. CEL, NOS remains a very aggressive disease with limited and non-standardized therapeutic options. HU or IFN can be used to control CEL-associated myeloproliferative features and steroids can reduce PB eosinophilia. In young and fit patients, alloSCT remains the only curative option [95].

In M/LN-Eo-TK, constitutive TK signaling is a potential target for TKIs, yet the efficacy of these compounds is highly variable. Of note, evidence on TKI therapy in such disorders relies only on single case reports and small case series. Despite this, the 2021 NCCN guidelines recommend TKIs as frontline treatment for both CP and BP disease, especially in the absence of clinical trials [96]. The type of gene fusion dictates response to TKIs. *PDGFRA* and *PDGFRB* translocations have shown exquisite sensitivity to imatinib [97,98]. 2G/3G-TKIs or *FLT3* inhibitors should be restricted to cases with primary/secondary imatinib resistance, as a bridge to alloSCT [99,100]. By contrast, *FGFR1*, *JAK2*, *FLT3* and

*ETV6::ABL1* rearrangements have more aggressive clinical course and benefit from other therapies (especially alloSCT) [101].

In M/LN with *FGFR1* rearrangement, sustained molecular remissions can be obtained with alloSCT or (more recently) with the selective *FGFR* inhibitor, pemigatinib [102,103]. In M/LN with *ETV6::ABL1* rearrangement, durable remissions have been observed with 2G/3G-TKIs in CP disease, with minimal (if any) effect in BP disease [101]. In M/LN with *FLT3* rearrangement, FLT3 inhibitors (sunitinib and sorafenib) have shown at best short-lived remissions and alloSCT represents the only curative option [104–106]. The same holds true for M/LN with *JAK2* rearrangement, for which only short-term remissions or a role in maintenance therapy have been reported for *JAK2* inhibitors (e.g., ruxolitinib) [101,107,108]. In conclusion, the treatment of M/LN-Eo-TK is still largely empirical and anecdotal. Further studies will hopefully disclose new molecular targets for such aggressive MNs.

## 6. Conclusions

In addition to morphology, immunophenotyping and cytogenetics (which have been the mainstay of diagnostic assessment over the past several decades and remain relevant today), genomic characterization and advances in sequencing technology have improved our understanding of MNs, opening novel therapeutic opportunities [109]. The need for genomic integration has prompted the formulation of new disease classification proposals, some of which have been discussed in this review. All of this certainly represents a challenge for pathologists and hematologists dealing with myeloid leukemias. Nevertheless, the sheer amount of data at our disposal represents an unprecedented opportunity to understand these neoplasms and, ultimately, our role in diagnosing and treating them.

Although the relevance of molecular tests cannot be overestimated, the 2022 ICC and WHO Classifications clearly demonstrate that integration of clinical, morphologic, immunophenotypic and genetic data remains the mainstay for the diagnosis of MNs [9,10]. This is in keeping with the decade-long approach to the diagnosis of hematopoietic disorders [94,110,111], which has shaped current classifications and treatment options. On the other hand, the discovery of recurrent genetic abnormalities and molecular determinants remains crucial for providing new therapeutic options.

In the foreseeable future, clinical–pathological and genetic studies will shed light on the still many questions concerning myeloid leukemias. The most compelling issues specifically regard (i) the relevance of blast thresholds for the diagnosis and prognosis of some MNs, (ii) the hierarchy of clinical, morphological and genetic data to define specific disease entities, (iii) the nature and type of molecular derangements to be included in genetically based classifications, (iv) the management of MNs characterized by poor response to conventional therapies and (v) the treatment of newly discovered entities. These problems can be solved only by the close collaboration of experts from different fields, in keeping with the inspiring principles of current hematology practice.

**Author Contributions:** M.P. and C.G.: writing of the paper; A.O.: review conceptualization, manuscript supervision and final editing. All authors have read and agreed to the published version of the manuscript.

**Funding:** This research received no external funding.

**Institutional Review Board Statement:** Not applicable.

**Informed Consent Statement:** Not applicable.

**Data Availability Statement:** Not applicable.

**Conflicts of Interest:** The authors declare no conflict of interest.

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
