# Peer review of "What’s New in the Classification, Diagnosis and Therapy of Myeloid Leukemias"

_hemato, doi:10.3390/hemato4020011_

Round 1

Reviewer 1 Report

Authors submitted a review on the different classification published in 2022 and made a special focus in four different myeloid malignancies . They underline differences between all these classifications and what will be pertinent for a clinical use. They use tables with colonums to underlin differences. Figure 1 is of particular interest to understand the way how to classify AMLs

Not sure that the histological pictures (Figure 2 and 3) are of a great interest in this article. May be it will be of interest to have a little table to uderline the different types of M/LN-Eo TK with specific molecular features. 

Author Response

ANSWERREVIEWER #1

Authors submitted a review on the different classification published in 2022 and made a special focus in four different myeloid malignancies. They underline differences between all these classifications and what will be pertinent for a clinical use. They use tables with columns to underline differences. Figure 1 is of particular interest to understand the way how to classify AMLs. Not sure that the histological pictures (Figure 2 and 3) are of a great interest in this article. May be it will be of interest to have a little table to underline the different types of M/LN-Eo TK with specific molecular features.

We thank the Reviewer for appreciating our review. To help readers with limited experience in the histopathology of myeloid disorders, we inserted 2 histological figures in our original submission, which were appreciated by Reviewer #2. For this reason, if the Reviewer agrees, we would keep them in the final version of the paper. We are grateful to the Reviewer also for appreciating Figure 1, which was specifically intended to help sub-classification of AML according to the ICC. To better illustrate differences in the classification framework of ICC and WHO, we added a further Figure presenting the hierarchical approach to the diagnosis of AML according to 2022 WHO (revised Figure 2). In keeping with the Reviewer’s request, we also added a Table listing the M/LN-Eo-TK according to the new classifications (Table 9).

Reviewer 2 Report

The authors present an excellent review focusing on the new features of both the ICC and WHO 2022. They have also managed to highlight in a synthetic and clarifying way the background of the changes with respect to WHO 2017 and the main differences between the new ICC and WHO 2022. In addition, the tables and figures enrich the article and allow to consult the new diagnostic criteria in an accurate way. 

I am very grateful for the work of the authors. It has been a pleasure to review the article. For my part, the article can be accepted in its present form. I have detected only two small typographical errors:

1) "moncytic leukemia"; page 2, line 41.

2) "thermobocytopenia"; page 9, line 330.

Author Response

ANSWER TO REVIEWER #2

The authors present an excellent review focusing on the new features of both the ICC and WHO 2022. They have also managed to highlight in a synthetic and clarifying way the background of the changes with respect to WHO 2017 and the main differences between the new ICC and WHO 2022. In addition, the tables and figures enrich the article and allow to consult the new diagnostic criteria in an accurate way. I am very grateful for the work of the authors. It has been a pleasure to review the article. For my part, the article can be accepted in its present form. I have detected only two small typographical errors:

1) "moncytic leukemia"; page 2, line 41.

2) "thermobocytopenia"; page 9, line 330.

We thank the Reviewer for appreciating our work and do apologize for the typing errors that have been fixed in the revised version of the paper.

Reviewer 3 Report

Summary: The authors summarize major changes in the classification of myeloid neoplasms arising from two classification systems that were revised in 2022, particularly focusing on key changes in AML, CMML, CML and myeloid neoplasms with eosinophilia.  Overall this was a well-written summary that captures most key points, at least within the disease entities which were chosen to focus on, with some changes in MDS and other diseases not included.  Notably, at times it seems there is more attention paid to the ICC system than the WHO system, with both being highly relevant.  

Focus on disease classification, though there is a basic overview of treatment of each of the diseases which is quite brief.

Comments:

-there is some commentary regarding blast thresholds in AML classification section, but I think warrants further discussion given change from historical thresholds. In particular, should include discussion of the MDS/AML category  recognized by ICC classification and relevance of this in treatment selection.

-There are changes in MDS classification in both schemes that would be worth discussing as well.

-Line 169; would also include appropriate selection of patients for allogeneic transplant/improving transplant outcomes as key area in AML treatment

-Line 177: should include the recently approved IDH1 inhibitor olutasidenib.

-Line 181: would add in caveat that CC-486 is indicated as a maintenance therapy.

-Line 193; should clarify that is for patients who underwent initial intensive therapy but were ineligible for HSCT

-Line 266: The wording here implies that there are genetically defined subsets within the ICC classification of CMML, which there are not. Is still important to note, though, that NPM1 can still classify as CMML as is subsequently detailed in the following lines.

-Within CMML, given this is new it is worth further highlighting the addition, in both systems, of flow/monocyte partitioning in the diagnostic criteria and specific features of this.

-Within the intro to section 3, in addition to commenting on JMML reclassification, the distinction of MDS/MPN-RS-T  with SF3B1 in new systems is also worth mentioning in the MDS/MPNs.

-Line 327: HMA/len combo has been studied for years now and hasn’t conclusively shown much efficacy, would suggest removing here.

-line 407: also aciminib here (same indications has ponatinib on FDA label; 2 prior TKIs or presence of T315I mutation for these two agents)

-under treatment of CML, briefly discuss treatment of blast phase CML as well, including role of allo-HSCT

-Line 488, would highlight the importance of allo-HSCT in non-PDGFRa/b diseases.

Author Response

ANSWER TO REVIEWER #3

Comments:

1) There is some commentary regarding blast thresholds in AML classification section, but I think warrants further discussion given change from historical thresholds. In particular, should include discussion of the MDS/AML category recognized by ICC classification and relevance of this in treatment selection.

We fully agree that blast threshold is one of the most relevant differences in the new classifications of AML and that the introduction of a new nomenclature for former MDS EB2 by the 2022 ICC (i.e. MDS/AML) is a turning point in MDS classification, with relevant clinical implications. We expanded this section in the revised version of the manuscript (end of paragraph 2.1).

2) There are changes in MDS classification in both schemes that would be worth discussing as well.

We thank the reviewer for this comment. As this invited review was specifically asked to focus on novelties in leukemias, we did not address MDS in our original submission. Nevertheless, we agree that AML and MDS are closely related entities (even more now, after the publication of the 2022 ICC and WHO Classifications). For this reason and to cope with the Reviewer’s request, we added a summary of the most relevant changes in MDS at the end of the AML section and produced a new Table presenting the ICC and 2022 WHO Classification of MDS (revised Table 2).

3) Line 169; would also include appropriate selection of patients for allogeneic transplant/ improving transplant outcomes as key area in AML treatment

We thank the Reviewer for this suggestion. The sentence has been modified accordingly.

4) Line 177: should include the recently approved IDH1 inhibitor olutasidenib.

Thank you. This information has been added in the revised version of the manuscript.

5) Line 181: would add in caveat that CC-486 is indicated as a maintenance therapy.

We thank the Reviewer for this suggestion. This information has been added in the revised version of the manuscript.

6) Line 193; should clarify that is for patients who underwent initial intensive therapy but were ineligible for HSCT

We agree that this point needed to be better specified and modified the sentence as per the Reviewer’s suggestion.

7) Line 266: The wording here implies that there are genetically defined subsets within the ICC classification of CMML, which there are not. Is still important to note, though, that NPM1 can still classify as CMML as is subsequently detailed in the following lines.

We thank the Reviewer for this comment. The sentence has been rephrased accordingly.

8) Within CMML, given this is new it is worth further highlighting the addition, in both systems, of flow/monocyte partitioning in the diagnostic criteria and specific features of this.

In the revised text (paragraph 3.2), we have added a sentence on the relevance of flow cytometry for the diagnosis of CMML, specifically stressing the role of ≥94% classical monocytes in PB for the differential diagnosis with other causes of monocytosis.

9) Within the intro to section 3, in addition to commenting on JMML reclassification, the distinction of MDS/MPN-RS-T with SF3B1 in new systems is also worth mentioning in the MDS/MPNs.

We agree that this is another relevant distinguishing feature of the new classifications. The original paragraph (introduction to section 3) has been modified to include also these novelties.

10) Line 327: HMA/len combo has been studied for years now and hasn’t conclusively shown much efficacy, would suggest removing here.

Following the Reviewer’s suggestion, we have removed HMA plus lenalidomide from this list.

11) Line 407: also asciminib here (same indications has ponatinib on FDA label; 2 prior TKIs or presence of T315I mutation for these two agents)

We thank the Reviewer for this comment. The sentence has been modified accordingly.

12) Under treatment of CML, briefly discuss treatment of blast phase CML as well, including role of allo-HSCT

The therapeutic options for blast phase CML have been expanded and better explicitated in the revised version of the manuscript.

13) Line 488, would highlight the importance of allo-HSCT in non-PDGFRa/b diseases.

Thank you. The relevance of allo-HSCT in M/LN-Eo-TK other than those with PDGFRA/B rearrangements has been emphasized in this sentence.